# Hypoglycemia Unawareness—A Review on Pathophysiology and Clinical Implications

**DOI:** 10.3390/biomedicines12020391

**Published:** 2024-02-08

**Authors:** Laura Hölzen, Bernd Schultes, Sebastian M. Meyhöfer, Svenja Meyhöfer

**Affiliations:** 1Institute for Endocrinology & Diabetes, University of Lübeck, 23562 Lübeck, Germany; laura.hoelzen@uksh.de (L.H.); bernd.schultes@friendlydocs.ch (B.S.);; 2Department of Internal Medicine 1, Endocrinology & Diabetes, University of Lübeck, 23562 Lübeck, Germany; 3Metabolic Center St. Gallen, friendlyDocs Ltd., 9016 St. Gallen, Switzerland; 4German Center for Diabetes Research (DZD), 85764 Neuherberg, Germany

**Keywords:** hypoglycemia unawareness, diabetes, hypoglycemia counterregulation, hormonal regulation, sleep deprivation

## Abstract

Hypoglycemia is a particular problem in people with diabetes while it can also occur in other clinical circumstances. Hypoglycemia unawareness describes a condition in which autonomic and neuroglycopenic symptoms of hypoglycemia decrease and hence are hardly perceivable. A failure to recognize hypoglycemia in time can lead to unconsciousness, seizure, and even death. The risk factors include intensive glycemic control, prior episodes of severe hypoglycemia, long duration of diabetes, alcohol consumption, exercise, renal failure, and sepsis. The pathophysiological mechanisms are manifold, but mainly concern altered brain glucose sensing, cerebral adaptations, and an impaired hormonal counterregulation with an attenuated release of glucagon, epinephrine, growth hormone, and other hormones, as well as impaired autonomous and neuroglycopenic symptoms. Physiologically, this counterregulatory response causes blood glucose levels to rise. The impaired hormonal counterregulatory response to recurrent hypoglycemia can lead to a vicious cycle of frequent and poorly recognized hypoglycemic episodes. There is a shift in glycemic threshold to trigger hormonal counterregulation, resulting in hypoglycemia-associated autonomic failure and leading to the clinical syndrome of hypoglycemia unawareness. This clinical syndrome represents a particularly great challenge in diabetes treatment and, thus, prevention of hypoglycemia is crucial in diabetes management. This mini-review provides an overview of hypoglycemia and the associated severe complication of impaired hypoglycemia awareness and its symptoms, pathophysiology, risk factors, consequences, as well as therapeutic strategies.

## 1. Introduction

### 1.1. Definition of Hypoglycemia

Hypoglycemia is a common complication in people with diabetes. It is defined by the American Diabetes Association as all episodes of an abnormally low plasma glucose concentration that expose the individual to potential harm [1]. In numerical terms, a value of 70 mg/dL (3.9 mmol/L) or lower with associated symptoms is usually considered a hypoglycemic condition. A blood glucose level below 70 mg/dL is also called level 1 hypoglycemia, defining the lower end of the postabsorptive glucose scale. Beyond that, hypoglycemia with glucose levels below 54 mg/dL (3.0 mmol/L) is called level 2 hypoglycemia, which is mostly accompanied by neurological symptoms of hypoglycemia. Level 3 hypoglycemia describes hypoglycemia independent of a defined numerical value as severe hypoglycemia requiring the assistance of a third party [2] (Table 1).

### 1.2. Symptoms of Hypoglycemia

Symptoms of hypoglycemia can be classified as autonomic and neuroglycopenic symptoms. While autonomic symptoms occur due to stimulation of the autonomic nervous system, neuroglycopenic symptoms are mainly caused by a glucose deficiency in the brain. Autonomic symptoms usually occur at higher glucose thresholds than neuroglycopenic ones [3,4]. Autonomic symptoms may include diaphoresis, palpitations, hunger, tingling, and anxiety. Neuroglycopenic symptoms include weakness, drowsiness, confusion and fatigue, seizures, and in the most severe cases may lead to coma and death.

### 1.3. Risk Factors for Hypoglycemia

Intensive glycemic control, often achieved through intensive insulin therapy, increases the risk of hypoglycemic episodes. This represents a major lifelong challenge for patients with diabetes because a low HbA1c level is important, but the risk of hypoglycemia increases with HbA1c values below 6.5% [5]. However, intensive insulin management has been shown to be a risk factor for a higher frequency of hypoglycemic episodes but not severe hypoglycemia [6].

Older age has also been shown to be related to severe hypoglycemic reactions [7]. Sociodemographic differences in risks for severe hypoglycemia were not found [6].

It is known that endogenous glucose release and production are decreased after alcohol ingestion and, thus, this represents a risk factor for severe hypoglycemia. During exercise, glucose utilization is generally increased and sensitivity to insulin is higher late after exercise and during the night as well as after weight loss. Especially in children and adolescents, nocturnal hypoglycemia can occur after delayed effects of exercise and alterations in sleep physiology. This is a serious complication and nocturnal hypoglycemic episodes are often profound and prolonged [8].

Hypoglycemia can also occur in certain medical conditions and diseases such as pancreatic or non-islet cell tumors, organ failure, dietary toxins, stress, and infections. Moreover, autoimmune conditions such as adrenal insufficiency and other endocrine diseases can lead to hypoglycemic episodes [9]. In patients with renal failure, for example, insulin clearance is reduced so that higher insulin levels could lead to severe hypoglycemia. Moreover, there is an impaired renal glucose production in these patients (Figure 1) [10].

After esophageal, gastric, or bariatric surgery, new onset of postprandial hypoglycemic episodes can occur. People with obesity who underwent Roux-en-Y gastric bypass are more affected than individuals who have undergone a sleeve gastrectomy. It has been shown that especially younger, female, and non-diabetic individuals experience hypoglycemic episodes more frequently after bariatric surgery [11]. The altered gastric anatomy allows undigested food to pass from the stomach into the small intestine too rapidly. Early dumping syndrome occurs within the first hour after food intake. Hyperosmolality of food causes a fluid shift into the intestinal lumen, leading to hypoglycemia with accompanying gastrointestinal symptoms. Late dumping syndrome typically occurs one to three hours after a meal. An incretin-driven hyperinsulinemic response to undigested carbohydrates can result in a hypoglycemic episode [12,13]. Elevated GLP-1 and associated insulin levels have been shown in patients after bariatric surgery [14]. Nevertheless, the risk of hypoglycemia during therapy with GLP-1 analogs is very low. It has even been shown that GLP-1 analogs can stabilize glucose levels. Therefore, the mechanism of late dumping syndrome appears to be more complex [13,15,16].

## 2. Pathophysiology of Hypoglycemia Unawareness Syndrome

### 2.1. Autonomic Failure

It is suggested that hypoglycemia unawareness syndrome is mainly induced by recent antecedent hypoglycemic episodes causing defective glucose counterregulation. The absence of symptoms of hypoglycemia reflects the attenuation of the sympathoadrenal response [17]. Hypoglycemia unawareness syndrome is commonly observed in individuals with recurrent hypoglycemic episodes disrupting the normal release of counterregulatory hormones, such as glucagon, growth hormone (GH), and epinephrine. Besides the attenuation of the sympathoadrenal responses, the blunted hormonal response leads to impaired glucose counterregulation and reduced awareness of hypoglycemia. Consequently, a vicious cycle of recurrent hypoglycemia with the potential for serious complications can occur [18,19]. Supporting the theory of hypoglycemia unawareness syndrome, consequent avoidance of hypoglycemia improves the hypoglycemic counterregulatory process and thus hypoglycemia awareness [18].

### 2.2. Altered Brain Glucose Sensing

The brain relies critically on glucose as its primary energy source. It is not possible for cells in the brain to synthesize glucose or to store it as glycogen. Therefore, brain cells are dependent on being constantly supplied with glucose. Due to the important glucose supply for the brain, specialized mechanisms can be activated by the brain to sense and respond to changes in glucose levels. These mechanisms involve various cell types, including neurons, glial cells, and endothelial cells, which work together to regulate glucose uptake, metabolism, and signaling. Dysfunction in these processes can lead to altered brain glucose sensing, which has far-reaching consequences on neuronal activity and overall brain physiology and can contribute to hypoglycemia unawareness syndrome [10,18].

Neurons express glucose transporters (GLUTs) that facilitate glucose entry into the cells. Moreover, certain neurons express glucose-sensitive channels, such as ATP-sensitive potassium (KATP) channels, which regulate neuronal excitability in response to glucose fluctuations. These glucose-sensing neurons are strategically located in key brain regions involved in energy homeostasis, such as the hypothalamus. Glial cells, particularly astrocytes, play a crucial role in brain glucose sensing. Astrocytes express GLUTs and actively take up glucose from the bloodstream. Intracellular glucose is metabolized through glycolysis, and the resulting lactate is released as a metabolic substrate for neurons. Astrocytes also release signaling molecules, such as ATP and lactate, which can modulate neuronal activity and synaptic transmission. Brain endothelial cells forming the blood–brain barrier (BBB) regulate the exchange of nutrients, including glucose, between the bloodstream and the brain. These cells express glucose transporters and glucose transport proteins (GLUT1 and GLUT3), ensuring a constant supply of glucose to the brain. Endothelial glucose-sensing mechanisms involve the release of signaling molecules, such as nitric oxide and lactate, which modulate blood flow and regulate glucose transport across the BBB. However, recent antecedent hypoglycemic episodes do not increase these mechanisms with a higher blood-to-brain glucose transport. Consequently, these mechanisms might work with a lower glucose threshold resulting in a higher risk of developing hypoglycemia unawareness syndrome [10,18,20].

### 2.3. Cerebral Adaptations

Recurrent hypoglycemic episodes can trigger adaptive responses within the central nervous system. Over time, the brain adapts to the lower glucose levels, resetting the threshold for glucose sensing to lower values. As a consequence, individuals with impaired hypoglycemia awareness exhibit a blunted response to falling glucose levels, further contributing to the condition [21].

Studies in animals have demonstrated the presence of glucose-sensing neurons in both the brain and peripheral regions, which detect changes in glucose concentration and transmit signals to initiate appropriate responses. These neurons can be either glucose-excited or glucose-inhibited, increasing or decreasing their activity based on glucose levels. Several neurotransmitters, including norepinephrine, gamma-aminobutyric acid (GABA), glutamate, and nitric oxide, are involved in relaying signals from these glucose-sensing neurons. The information on glucose concentration is integrated at the central level, primarily in the hypothalamus and hindbrain, and then transmitted to motor neurons responsible for autonomic and neuroendocrine responses. The precise roles of peripheral and central glucose sensors in initiating the counterregulatory response are still being studied [22]. Human studies have identified brain areas activated during hypoglycemia, including the hypothalamus, brainstem, anterior cingulate cortex, and others. An important mediator in the development of hypoglycemia unawareness syndrome could be levels of GABA, a potent inhibitory neurotransmitter. The concentration levels of GABA are decreased during an acute hypoglycemic episode in ventromedial hypothalamus (VMH) interstitial fluid. However, recurrent hypoglycemia induces increased GABA levels so that the absence of decreasing GABA concentrations during hypoglycemia is associated with a reduced glucagon and epinephrine response. Recurrent hypoglycemia results in increased VMH GABA inhibitory tone which could be important in the development of a hypoglycemia unawareness syndrome [21].

### 2.4. Hormonal Regulation

Hypoglycemia counterregulation is a complex process involving further counterregulatory hormones such as epinephrine, glucagon, cortisol, and other hormones. These hormones mostly work synergistically to increase hepatic glucose production, reduce peripheral glucose uptake, and mobilize alternative energy sources during hypoglycemia.

In healthy individuals without diabetes, the initial response to low blood sugar levels is a reduction in insulin secretion, even before the plasma glucose concentration reaches hypoglycemic levels. Glucagon and insulin play an important role in hypoglycemia counterregulation and in restoring normal glucose homeostasis. Hormonal dysregulation has been identified as a significant factor contributing to the development of hypoglycemia unawareness syndrome. In individuals with T1D or prolonged type 2 diabetes (T2D), unregulated insulin release from subcutaneous depots or the sustained effects of sulfonylureas can result in elevated systemic insulin levels during hypoglycemia. In addition, beta-cell dysregulation leads to dysfunction of alpha-cells, and a lack of a decrease in insulin levels prevents glucagon secretion. These processes cause defects in counterregulatory response, which is the failure to remove insulin from the systemic circulation. The relative insulin excess increases glucose uptake and suppresses glucose production in the liver, despite the development of hypoglycemia. It also acts peripherally to limit lipolysis and the release of gluconeogenic substrates to the liver. These effects collectively enhance tissue glucose uptake and reduce glucose production, intensifying the hypoglycemic stimulus. Furthermore, there is a paradoxical impairment of glucagon secretion during hypoglycemia in patients with diabetes. The reasons for this reversal are not fully understood, but it may involve the loss of regulatory signals from beta-cells, such as insulin and GABA [23,24,25]. In addition, the intraislet insulin hypothesis could play an important role. In this hypothesis, insulin secretion from β-cells decreases during hypoglycemia and, at the same time, serves as an activating signal for the release of glucagon from α-cells. A more recent hypothesis suggests that zinc atoms, and not the insulin molecule itself, inhibit α-cells and thus glucagon secretion via their ability to open ATP-sensitive K+ channels in α-cells [26].

GH is secreted at glucose concentrations of approximately 66 mg/dL (3.7 mmol/L) and is also involved in hypoglycemia counterregulatory processes. Some studies suggest that GH has a prominent role in the setting of a prolonged hypoglycemic episode. GH induces glucose level changes over several hours by stimulating lipolysis in adipose tissue as well as ketogenesis and gluconeogenesis in the liver [22,27,28]. For example, in patients with GH deficiency due to hypopituitarism, plasma glucose concentrations were significantly lower 12 h after continuous insulin infusion to induce hypoglycemia [29]. After antecedent hypoglycemia, GH responses were significantly suppressed [22].

Glucocorticosteroids increase when hypoglycemia occurs, which has been proposed to feedback to the hypothalamus. However, the exact role of cortisol in hypoglycemia counterregulation remains unclear. In contrast to GH, cortisol effects the central nervous system. A prior elevation of cortisol levels and recurrent hypoglycemia lead to an attenuation of counterregulatory response to subsequent hypoglycemia. Consequently, cortisol could play an important role in the development of hypoglycemia unawareness syndrome [30].

The concept of hypoglycemia unawareness also includes that recent antecedent hypoglycemic episodes cause a defective counterregulation by reducing epinephrine levels. Physiologically, in response to hypoglycemia, plasma levels of epinephrine increase. After recent antecedent hypoglycemia, epinephrine levels are significantly reduced (Figure 2) [19,30].

### 2.5. Sleep

Emerging evidence suggests that sleep plays a critical role in hypoglycemia counterregulation [19]. It is known that sleep quality and duration have a significant impact on metabolic as well as neuroendocrine processes [31,32,33]. Sleep is a complex physiological process characterized by distinct stages, including non-rapid eye movement (NREM) sleep and rapid eye movement (REM) sleep. Both stages contribute to the regulation of glucose metabolism. NREM sleep is associated with increased insulin sensitivity and glucose uptake in peripheral tissues. In contrast, REM sleep is characterized by a shift toward insulin resistance and increased glucose production by the liver. Disruptions in sleep architecture can disrupt glucose homeostasis, predisposing individuals to hypoglycemic episodes [33,34].

Hypoglycemia during sleep is particularly challenging as it often goes unnoticed, leading to prolonged and severe episodes. Studies have demonstrated that healthy individuals exhibit blunted autonomic and symptomatic responses to hypoglycemia during sleep compared to wakefulness [21]. Sleep-related changes in hormonal secretion, such as reduced release of counterregulatory hormones (e.g., glucagon, epinephrine, growth hormone), contribute to impaired hypoglycemia awareness during sleep. Around 60–70% of nocturnal hypoglycemic episodes in patients with diabetes occur during late sleep, i.e., between 03:00 and 07:00 h [35,36]. Due to a difference in sleep stage architecture, there is also a difference between hormonal counterregulation during early and late nocturnal sleep. It was shown that counterregulatory hormones such as epinephrine, norepinephrine, ACTH, cortisol, and growth hormone were less pronounced during late than early nocturnal hypoglycemia [37]. In addition, epinephrine response to early nighttime hypoglycemia was shown to be enhanced, which is important in mediating the awareness of hypoglycemia [38]. Awakening from sleep due to hypoglycemia is a mechanism which is mainly part of the central nervous system. This was shown by a study with patients with T1D and healthy controls who underwent a hyperinsulinemic–hypoglycemic glucose clamp with a glucose nadir of 40 mg/dL (2.2 mmol/L) during sleep. Awakening from sleep with an accompanying increase in epinephrine levels was impaired in patients with T1D compared to healthy controls [39]. Furthermore, it is known that sleep represents a potential modulator of metabolic memory. In a study with two hypoglycemic episodes, followed by a third hypoglycemic clamp after one night of regular sleep or sleep deprivation, the hormonal counterregulatory response to hypoglycemia was preserved after sleep loss. In contrast, counterregulatory response was attenuated after regular sleep, which can be considered as a learning process of memory consolidation that implicates the formation of neurometabolic memory and contributes to the development of a hypoglycemia unawareness syndrome [19,40].

### 2.6. Consequences

Hypoglycemia unawareness poses significant safety risks to patients. Individuals experiencing hypoglycemia unawareness are more likely to undergo severe hypoglycemic episodes, leading to accidents, injuries, and increased healthcare utilization. Moreover, the fear and anxiety associated with hypoglycemia can impair quality of life, causing emotional distress, social isolation, and reduced adherence to treatment regimens. In addition, patients with recurrent hypoglycemia are more likely to suffer from depression and anxiety [21].

In patients with T1D, cognitive function and intellectual activity are likely to be impaired during acute hypoglycemia. Furthermore, this impairment persists for a longer period after a hypoglycemic episode [41]. Supporting this fact, imaging techniques showed that people with T1D require a higher level of brain activation to attain the same level of cognitive performance during hypoglycemia compared with healthy individuals [42]. However, some studies such as the Diabetes Control and Complications Trial [43] and the Stockholm Diabetes Interventions Study [44] could not detect any cognitive dysfunctions after recurrent hypoglycemia [21]. Concerning older adults, a large longitudinal cohort study revealed an increased risk for dementia in older individuals with T2D when having a history of severe hypoglycemic episodes [45]. Moreover, older individuals are at higher risk for geriatric syndromes including frailty, cognitive impairment, and depressive symptoms [21].

Episodes of severe hypoglycemia are associated with an increased risk of mortality. Although an intensive diabetes therapy is important in preventing other diabetes complications, hypoglycemia should always be avoided. Hypoglycemia induces vascular effects such as the activation of prothrombotic, proinflammatory, and proatherogenic components resulting in an increased risk for cardiovascular diseases [46,47]. Another study found an association of hypoglycemia and acute cardiovascular events such as myocardial infarction, particularly in patients who experience considerable swings in blood glucose [48].

Especially during pregnancy, severe hypoglycemia and hypoglycemia unawareness occur up to five times more frequently in women with diabetes, especially during the first trimester. Risk factors for severe hypoglycemia in pregnant women are a long duration of diabetes, an HbA1c level ≤ 6.5%, and high doses of daily insulin [49]. Consequences of severe hypoglycemia during pregnancy are an impairment in fetal lung maturation, intellectual performance and psychomotor development, and fetal growth [50].

## 3. Therapeutic Strategies

Hypoglycemia unawareness is a challenging condition due to the impaired ability to recognize and respond to low blood glucose levels. The primary goal to prevent hypoglycemia unawareness should be the avoidance of hypoglycemic episodes. It has been shown that even two to three weeks of strict avoidance of hypoglycemia in patients with hypoglycemia unawareness is helpful to increase the glucose threshold triggering the onset of hypoglycemic symptoms to a healthy level [51]. Therefore, providing comprehensive education to individuals with hypoglycemia unawareness and their caregivers is crucial. Empowering patients with knowledge about hypoglycemia management, including recognition and appropriate treatment, can enhance their ability to self-manage and prevent severe events [21]. A randomized, prospective multi-center study found that specific training programs for patients with hypoglycemia unawareness issues provided additional benefits compared to the control group. These benefits included improving hypoglycemia awareness, reducing mild hypoglycemia episodes, and enhancing the detection and treatment of low blood glucose levels [52]. Another study, which focused on assessing the restoration of impaired hypoglycemia awareness and defective hypoglycemia counterregulation through educational strategies over 6 months, observed a significant increase in the mean glucose concentration at which participants first experienced symptoms of hypoglycemia compared to baseline. Additionally, the participants exhibited enhanced counterregulatory responses to hypoglycemia [53]. These findings emphasize the importance of educational programs and specific training interventions in improving hypoglycemia awareness and management.

There is recent evidence that sleep is metabolically relevant and plays a crucial role in the adaptation process of the counterregulatory response to hypoglycemia. The adaptation to recurrent hypoglycemia leading to hypoglycemia unawareness has been described as a learning process that implicates the formation of neurometabolic memory. In a recent study, sleep deprivation compared with sleep attenuated the adaptation to recurrent hypoglycemia. The counterregulatory response of the key hormones epinephrine, GH, and glucagon was dampened after recurrent hypoglycemia when participants slept regularly compared to sleep deprivation. Furthermore, neuroglycopenic symptoms during hypoglycemia were preserved upon sleep deprivation [19]. Although sleep loss or a short sleep duration has negative effects on glucose metabolism, appetite, and hunger [31,32,33], sleep loss after repeated hypoglycemia could prevent hypoglycemia unawareness. However, further investigations about sleep-related memory formation in chronic hypoglycemia unawareness are needed.

Furthermore, improving glycemic control with individualized glycemic targets for blood glucose control is essential to reduce hypoglycemic episodes. Any comorbidities, such as cardiovascular disease, history of severe hypoglycemia, hypoglycemia unawareness, low glycated HbA1c, low C-peptide levels, or autonomic neuropathy, should always be considered. Individuals with older age and/or frailty should not reach the standard euglycemia or HbA1c levels <7.5%. Avoiding hypoglycemic episodes should be a priority in elderly patients [54]. The insulin treatment should also be individually adapted for each patient. To avoid severe hypoglycemic episodes, in many cases it could be helpful to use a rapid-acting insulin analog instead of regular insulins. Rapid-acting insulins have a faster onset of action, a higher peak, and shorter duration of action, which corresponds more closely to the endogenous mealtime insulin response. This allows more flexibility in terms of meals and exercises [55,56]. NPH insulins should be used with caution due to their ability to trigger hypoglycemia more frequently. In contrast, long-acting insulin analogs show a more consistent, longer, and flatter action profile [10]. With the latest generation of insulin analogs such as degludec and glargine U300, it is easier to reach a consistent action profile [56]. Compared to insulin glargine U100, insulin degludec could show a reduced rate of hypoglycemic episodes [57]. For patients with T2D, sulfonylureas should only be used with caution and should be avoided in elderly individuals due to the higher risk of hypoglycemia. Rates of hypoglycemia are higher especially for glibenclamid than for other sulfonylureas [58]. Other treatment options for patients with T2D such as metformin, glucagon-like peptide 1 receptor agonists, dipeptidyl peptidase-4 inhibitors, pioglitazone, and sodium-glucose co-transporter-2 inhibitors show in contrast beneficial effects in reducing the risk of hypoglycemia [56].

Historically, self-monitored blood glucose (SMBG) was crucial in the prevention of hypoglycemia during intensified insulin therapy. It is known that a high frequency of SMBG monitoring (≥10 times per day) is associated with a better glucose control in all age groups [59]. However, for many people, it is not possible to practice SMBG monitoring with such frequency. Continuous glucose monitoring (CGM) devices that measure glucose concentration in the interstitial fluid have revolutionized diabetes management. These devices, typically non-invasive or minimally invasive, use a sensor inserted under the skin to measure glucose levels continuously and transmit the data to an external monitor. CGM devices offer a wealth of glucose data, allowing for better management of diabetes and therapy. They provide real-time and continuous glucose level monitoring, enabling the detection, correction, and prevention of hyper- or hypoglycemic events. CGM devices can display current glucose levels and trends, generate alerts for hypoglycemia or hyperglycemia, and aid in optimizing treatment decisions [56]. Real-world evidence data on CGM management in adult individuals with T1D or T2D showed significantly lower severe hypoglycemic event rates after <6 months with CGM [60]. The combination of CGM with continuous subcutaneous insulin infusion (CSII) has led to further technology development. The combination of CGM and CSII is particularly widespread in children and adolescents with a use rate of >90% in countries with optimal reimbursement [61]. With this combination, it is possible to shift HbA1c values and “time in range” (TIR) into the normal range without increasing the risk of hypoglycemia [62]. The newer technology, “automated insulin delivery” (AID) or “hybrid closed-loop system” (HCL), is equipped with control algorithms that can automatically suspend insulin infusion to prevent hypoglycemia. This technology is associated with a reduced risk of hypoglycemia and a higher TIR [63]. The current hybrid closed-loop systems require the manual entry of carbohydrates consumed to calculate prandial doses [64]. Closed-loop systems, also known as artificial pancreas systems, combine continuous glucose monitoring (CGM) and insulin delivery without requiring patient intervention. These systems utilize a control algorithm that adjusts insulin infusion frequently (i.e., every 5 min) based on glucose levels and data from CGM [65]. It has been shown that closed-loop systems reduce time spent in hypoglycemia [56,66].

## 4. Conclusions

In conclusion, hypoglycemia unawareness is a significant complication in diabetes treatment that poses serious risks to individuals. Failure to identify hypoglycemia in a timely manner can lead to severe consequences, including unconsciousness, seizures, and even death. Various risk factors contribute to the development of hypoglycemia unawareness, such as intensive glycemic control, prior episodes of severe hypoglycemia, longer duration of diabetes, alcohol consumption, exercise, renal failure, and sepsis. The pathophysiology of hypoglycemia unawareness involves multiple mechanisms, including altered brain glucose sensing, cerebral adaptations, and impaired hormonal counterregulation. These processes disrupt the normal response to hypoglycemia, leading to a blunted release of counterregulatory hormones such as glucagon, epinephrine, and growth hormone. The impaired hormonal counterregulatory response can initiate a vicious cycle of recurrent hypoglycemia and further exacerbate the condition. Additionally, there are alterations in brain glucose sensing, where the brain fails to detect and respond appropriately to low glucose levels.

Prevention of hypoglycemia is crucial in managing diabetes and addressing hypoglycemia unawareness. Avoiding hypoglycemic episodes and maintaining stable blood glucose levels can help to restore hypoglycemia awareness. Therapeutic strategies involve a comprehensive approach, including medication adjustments, dietary modifications, regular physical activity, and continuous glucose monitoring. Additionally, education and awareness about hypoglycemia symptoms and proper self-care practices are essential for individuals with diabetes.

In summary, hypoglycemia unawareness represents a significant challenge in diabetes management, but with appropriate preventive measures and therapeutic strategies, the risks associated with this condition can be minimized. Further research is needed to gain a deeper understanding of the underlying mechanisms and develop targeted interventions to improve hypoglycemia awareness in individuals with diabetes.

## Figures and Tables

**Figure 1 biomedicines-12-00391-f001:**
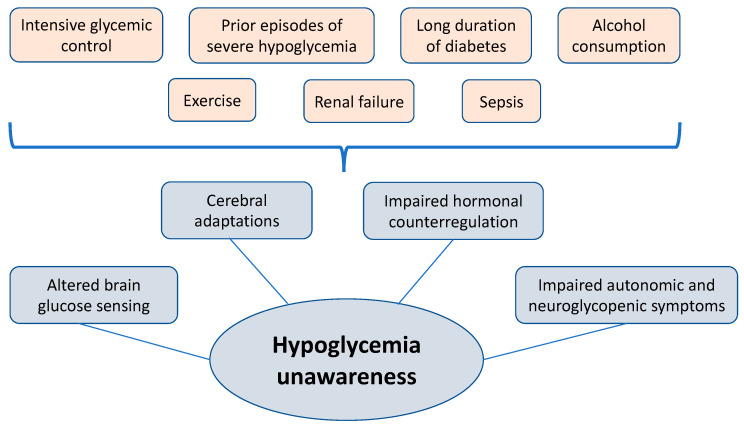
Graphical representation of the risk factors for hypoglycemia unawareness and its pathophysiology.

**Figure 2 biomedicines-12-00391-f002:**
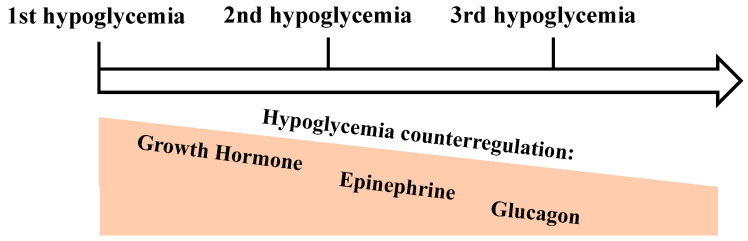
Concept of a dampened hormonal hypoglycemic counterregulation after recent antecedent hypoglycemic episodes.

**Table 1 biomedicines-12-00391-t001:** Definition of levels of hypoglycemia [2].

Level	Criteria
**Level 1**	Plasma glucose concentration < 70 mg/dL (<3.9 mmol/L); ≥54 mg/dL (≥3.0 mmol/L)
**Level 2**	Plasma glucose concentration < 54 mg/dL (<3.0 mmol/L)
**Level 3**	A severe event characterized by altered mental and/or physical status requiring assistance

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
