# Peer review of "Hypoglycemia Unawareness—A Review on Pathophysiology and Clinical Implications"

_biomedicines, 2024, doi:10.3390/biomedicines12020391_

Round 1

Reviewer 1 Report

Comments and Suggestions for Authors

Authors have summarized the available knowledge about a clinically importent thing. Hypoglicemia can also be present in diabetic patients causing even serious consequences. The mini-review describes the definition of hypoglycemia, its different regulatory and influencing factors. At the end we can obtain futher insight into the possible therapeutical strategies, which are for avoiding the recurrent episodes of hypoglycemia. 

The manuscript is well-written, provides enough information about this topic. In summary, this paper can be accepted in its present form.

Author Response

We thank the reviewer for this positive feedback!

Reviewer 2 Report

Comments and Suggestions for Authors

The authors present a very comprehensive review on hypoglycemia unawareness. It was so comprehensive in physiology  that the actual point of the paper almost got lost to it. I'd recommend that the authors take more time to summarize the physiology to make it shorter and perhaps end each section of the physiology with how it contributes to hypoglycemia unawareness. For example the chapter on sleep had much on how sleep contributes to insulin and glucose metabolism, but quite little how it contributes to hypoglycemia unawareness.

Author Response

We would like to thank the reviewer for the constructive comments to improve the manuscript.

  • We shortened the sections on physiology and mentioned at the end in each section how it contributes to hypoglycemia unawareness.
  • We shortened the section on sleep (and the contribution on insulin and glucose metabolism).
